# Long-Term Conservation Tillage Increases Yield and Water Use Efficiency of Spring Wheat (*Triticum aestivum* L.) by Regulating Substances Related to Stress on the Semi-Arid Loess Plateau of China

Changliang Du [1], Lingling Li [1,*], Junhong Xie [1], Zechariah Effah [1,2], Zhuzhu Luo [1] and Linlin Wang [1]

[1] State Key Laboratory of Aridland Crop Science, College of Agronomy, Gansu Agricultural University, Lanzhou 730070, China; changliangdu111@163.com (C.D.)

[2] CSIR-Plant Genetic Resources Research Institute, Bunso P.O. Box 7, Ghana

* Correspondence: lill@gsau.edu.cn; Tel.: +86-0931-7631145

**Abstract:** Improving the water-use efficiency (WUE) of crops is the most effective way to increase yields in semi-arid regions. Field research was carried out based on a long-term experiment initiated in 2001, aimed to explore the mechanisms of different tillage practices effects on grain yield and WUE of spring wheat. Tillage practices in the research including conventional tillage (CT), no tillage with no straw mulching (NT), conventional tillage with straw incorporation (TS), and no tillage with straw mulching (NTS). The effects of tillage practices on soil's physical and chemical properties, dry matter accumulation, grain yield, dynamics of stress-related substances, and WUE were observed. Soil and plant samples in this research were collected in 2020 (wet year), 2021 (dry year), and 2022 (dry year). The results indicated that NTS improved the soil's physical and chemical properties. The NTS treatment had the lowest soil bulk and pH and the highest total N, $NO_3^-$-N, and available P. Throughout the whole growth stage, soil water content in the NTS and TS treatments were significantly higher than that of CT by 8.77–20.40% and 2.19–18.83, respectively. Averaged catalase (CAT), peroxidase (POD), and soluble protein across the three years with NTS and TS were significantly increased by 1.26–25.52% compared to CT. Meanwhile, the NTS treatment had the lowest malondialdehyde (MDA) content among the different tillage practices. NTS maintained the highest dry matter accumulation throughout the whole growth stage among different treatments; it was increased by 10.47–73.33% compared with CT. The average grain yields and WUE of NTS across the three years were 6.09–30.70% and 6.79–40.55% higher than other tillage practices, respectively. It is concluded that NTS influences dry matter accumulation and water-use efficiency during the whole growth stage of spring wheat by improving the soil's physicochemical properties and modulating spring wheat substances related to stress, which in turn promotes yield formation.

**Keywords:** spring wheat; tillage practices; substances related to stress; yield

## 1. Introduction

The Loess Plateau is an important grain-producing region in China; however, it is severely affected by soil erosion and other environmental factors [1]. Wheat (*Triticum aestivum* L.) is a major food crop in the region. The low and concentrated rainfall and arid climate in the region limit the development of agriculture [2,3], and the growth of spring wheat in the region is dependent on erratic precipitation. Conservation agriculture [4,5] is considered a reliable agricultural management practice in the region. Previous studies have shown that conservation tillage can suppress weed growth [6], reduce wind erosion [7], improve the physical properties of the soil [8], and increase rainwater-use efficiency [9,10], thus ensuring higher crop yields. The application of conservation measures on the Loess Plateau can improve precipitation-use efficiency and, thus, bring crops

close to their potential yield [11,12]. Crop yield increase depends not only on climatic conditions but also on crop stress tolerance. Understanding the external environmental and physiological mechanisms regulating crop growth and yield under these conditions is of great importance [12].

Substances related to stress are an important component in effecting the plant growth and yield. Changes in stress-related substances within the crop affect the growth and development of the crop, which in turn affects the yield [13,14]. Crops usually respond to drought-induced oxidative stress by regulating substances related to stress [15,16]. Proline, soluble protein content, catalase (CAT) and peroxidase (POD) are helpful to maintain the balance of reactive oxygen species and improve the resistance of crops [17,18]. The rate of MDA accumulation in the plant can effectively reflect the strength of the plant's tissue scavenging ability of free radicals, and the higher the MDA content in the crop, the greater the damage [19,20]. CAT can directly break down $H_2O_2$ into molecular oxygen and water and POD can further scavenge hydrogen peroxide ($H_2O_2$) by oxidizing the substrate [21]. Some studies have shown that proline content CAT, and POD activity were higher under the straw mulching treatment than under the no mulching treatment [22] and that conservation tillage can significantly increase antioxidant enzyme activity in the crop [23]; however, it was also shown that field management practices did not have a significant effect on the antioxidant enzyme system in crops [24]. It has also been shown that conservation tillage can alter antioxidant enzyme activity in crops [25,26]. The function of anti-stress substances in the crop is well established, but the content and activity of anti-stress substances in the crop under different tillage practices is still controversial. The study of the effects of different tillage practices on crop substances related to stress is essential for a comprehensive understanding of plant growth, water-use efficiency, and yield.

Therefore, based on a long-term conservation tillage trials initiated in 2001, this study plans to detect the effects of tillage practices on soil physicochemical properties, crop substances related to stress, and their relationship with crop dry matter accumulation, yield, and WUE, with the aim to explore the mechanisms of different effects of tillage practices on grain yield and WUE of spring wheat.

## 2. Materials and Methods

### 2.1. Site Description

The trial site is located at the Gansu Agricultural University's Rainfed Agricultural Experimental Station (35°28′ N, 104°44′ E) in Lijiabao Town, Dingxi City, Gansu Province, China. The soil and plant samples in this study in 2020–2022 were taken from the long-term locational conservation tillage experiment established in 2001. The test site belongs to the arid region of the Loess Plateau with an average altitude of 2000 m and a frost-free period of 140 days. Test land is a calcaric cambisol [27], medium and low fertility. The annual cumulative temperature >10 °C is 2240 °C; the annual radiation is radiated from 5930 m MJ$^{-2}$ and the sunshine is 2480 h. The precipitation during the spring wheat growth stage (March–July) in 2020–2022 was 291.6, 168.6, and 121.9 mm, respectively. The specific precipitation situation is shown monthly in Figure 1. The physicochemical properties of the soil in the experiment during 2020–2022 were shown in Table 1.

**Table 1.** Soil physical and chemical properties in the field during 2020–2022.

| Treatments | Bulk Density (Mg m$^{-3}$) | pH | Total N (g kg$^{-1}$) | NO$_3$-N (g kg$^{-1}$) | Available P (mg kg$^{-1}$) | Available K (mg kg$^{-1}$) |
|---|---|---|---|---|---|---|
| conventional tillage (CT) | 1.28 ± 0.02 a | 8.37 ± 0.01 a | 0.77 ± 0.01 b | 23.62 ± 0.60 b | 13.63 ± 0.18 b | 314.80 ± 39.92 a |
| no tillage with no straw mulching (NT) | 1.25 ± 0.03 ab | 8.33 ± 0.02 b | 0.78 ± 0.02 b | 23.60 ± 0.89 b | 13.17 ± 0.63 b | 323.07 ± 45.89 a |
| conventional tillage with straw incorporation (TS) | 1.23 ± 0.02 b | 8.36 ± 0.00 a | 0.80 ± 0.02 a | 25.21 ± 0.53 a | 14.12 ± 0.20 ab | 318.67 ± 40.29 a |
| no tillage with straw mulching (NTS) | 1.18 ± 0.01 c | 8.29 ± 0.02 b | 0.82 ± 0.03 a | 25.56 ± 0.40 a | 14.63 ± 0.65 a | 333.17 ± 37.07 a |

Note: Within a column for a given year, means followed by different letters are significantly different ($p \leq 0.05$). Same as below.

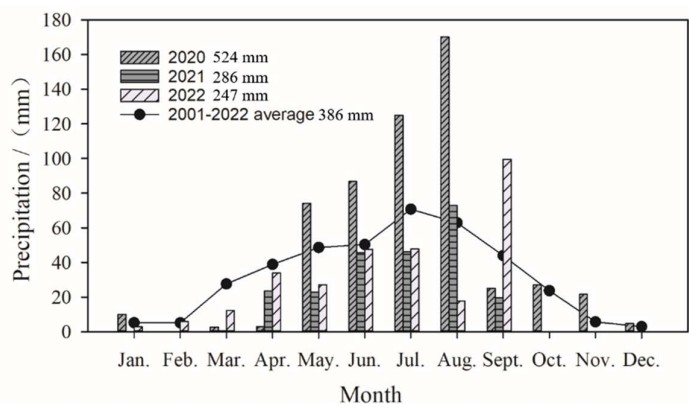

**Figure 1.** Monthly total precipitation for January through December in 2020, 2021, 2022, and the 2001–2022 average at the study site.

## 2.2. Experiment Design

There were four treatments, and three replications were set for each treatment, including conventional tillage (CT): after harvest, the field was ploughed three times and harrowed twice, and all stubbles were removed after harvesting the crop; no tillage with straw mulching (NTS): the field was left un-tilled throughout the experiment, and a no-till planter was used for the application of fertilizers and seed sowing, and all the straw of previous crop was returned to and mulched the original plot; no tillage with no straw mulching (NT): the field was left un-tilled throughout the experiment. Sowing and fertilization were performed with seedling-machine at the same time; and conventional tillage with straw incorporation (TS): the field was ploughed and harrowed exactly as CT treatment, but with straw incorporated at the first plough. All the straw of the previous crop was returned to the original plot immediately after threshing and then incorporated into soil. Field management and specific treatment measures are described by Huang et al. [28]. A double sequence rotation cropping pattern of spring wheat and peas with interannual rotation of the two crops was used. The spring wheat and pea varieties are "Dingxi 40" and "Lvnong 2", respectively. "Dingxi 40" is a national appraisal variety (National Appraisal Wheat 2009032), with medium drought resistance [29]. Spring wheat was planted in late March and harvested in late July each year; peas were planted in early April and harvested in early July each year. The seedling rate was 187.5 kg ha$^{-1}$ for spring wheat in rows spaced 20 cm apart and 180 kg ha$^{-1}$ for peas in rows spaced 24 cm apart. Calcium superphosphate (105 kg $P_2O_5$ ha$^{-1}$ for wheat and peas) and urea (105 and 20 kg N ha$^{-1}$ for wheat and peas, respectively) were applied to the soil with the seeds at planting. Each experimental plot was randomly distributed, and each plot was 20 × 4 m = 80 m$^2$. All samples for this study were collected in the spring wheat field. Weeds were removed by hand during the growing season and controlled with herbicides during the fallow period.

## 2.3. Sampling and Measurements

### 2.3.1. Soil Physical and Chemical Properties

Soil moisture content was determined using the oven-drying method [30] in the 0–5, 5–10, and 10–30 cm soil layers at each fertility period of spring wheat, and the soil moisture content in the 30–50, 50–80, 80–110 cm soil layers per plot was determined using a time-domain reflectance soil moisture sensor (TRIME-PICO, IPH/T3, IMKO GmbH, Ettlingen, Germany). Soil physical and chemical properties were sampled before sowing of spring wheat in 2020–2022. Soil bulk density in the 0–5, 5–10, and 10–30 cm soil layers was determined using the ring-knife method [30], with the 0–30 cm soil layer as its mean value.

Three soil cores (4.5 cm inner diameter) were taken randomly within each plot from the 0–5, 5–10, and 10–30 cm soil depths before sowing of spring wheat in 2015 for measurement of soil nutrient contents. In each plot, three soil cores (4.5 cm inner diameter) were taken randomly from the 0–5, 5–10, and 10–30 cm soil depth, brought back to the laboratory,

air-dried, ground, and passed through a 2.0 mm sieve. Soil total nitrogen (Total N) content was determined using the method of Semimicro, Kjeldahl. et al. [31], using concentrated sulfuric acid and a catalyst, followed by a fully automatic Kjeldahl nitrogen tester (Model A1225, Königswinter, Germany). Additionally, using the Cavagnaro, T. et al. [32] for the determination of soil nitrate nitrogen (NO3–N). Available phosphorus (Available P) was determined using the Olsen method [33]. Available potassium (Available K) was determined using the 1M NH4Ac leaching-flame photometric method (M410, Sherwood, UK).

### 2.3.2. Stress-Related Substances

Samples were taken at the seedling, jointing, heading, flowering, and filling stages of spring wheat in 2020–2022. Fifteen spring wheat leaves were randomly taken in each plot, wrapped in aluminum foil, and immediately placed in a bubble box containing liquid nitrogen, which was brought back to the laboratory and transferred to a $-80\,^{\circ}\text{C}$ refrigerator for measurement. Proline was determined using the acidic ninhydrin method [34]; soluble protein content was determined using the Komas Brilliant Blue method [34]; CAT was determined according to Sun [35]; and POD was determined using the method described by Rahnamal and Ebrahmzdeh [36]. MDA content was determined by the method of Sun & Hu [37].

### 2.3.3. Dry Matter Accumulation, Grain Yield, and WUE

Whole plant samples were taken at wheat seedling, jointing, heading, flowering, and filling and grouped according to the experimental plots. Five spring wheat plants were randomly measured from each plot for determination of total aboveground dry matter. The tissue was placed in an oven, dried at $105\,^{\circ}\text{C}$ for 30 min, dried at $75\,^{\circ}\text{C}$ to constant weight, and then weighed. At maturity, each plot's edges (0.5 m) were removed, and all remaining plots were harvested by hand with a sickle, threshed using a thresher, and weighed.

### 2.3.4. Statistical Analysis

The analysis of variance (ANOVA) test was used to evaluate soil moisture content, wheat dry matter accumulation, and the substances related to stress. Tillage practices were considered as fixed effects; years, replication, and precipitation were considered random effects. SPSS 19.0 (Company, Chicago, IL, USA) statistical software was used to detect differences among treatments at the 0.05 significance level with the LSD test. Statistical analysis and data graphing were performed with Excel 16.0 and SigmaPlot 14.0, respectively.

## 3. Results

### 3.1. Soil Physical and Chemical Properties

The soil's physical and chemical properties were significantly affected by tillage methods (Table 1). During 2020–2022, NTS had the significantly lowest bulk density compared to the other treatments. In addition, NTS and TS had the significantly lowest pH and highest total N, $NO_{3-}$-N, and available P. There was no significant difference in available K under different tillage methods.

Soil water content varied with tillage methods, soil depth, and growth stage of spring wheat (Figure 2). Averaged across the three years, soil water content at the sowing stage with NTS and TS was significantly increased by 8.77–10.90% and 2.19–18.83% compared to CT in 0–80 cm soil depth, respectively. TS, NTS, and NT significantly increased soil water content by 20.40%, 13.71%, and 6.16% compared to CT in the 5–10 cm soil layer at the seedling stage, respectively. At the jointing and heading stage, NTS significantly increased soil water content by 11.88% and 11.48% compared to CT in the 0–5 cm soil layer, respectively. NT did not significantly improve soil water content compared to CT. CT had the lowest soil water content compared to other treatments at the flowering stage. NTS significantly increased soil water content by 16.91% and 10.38% compared to CT, respectively, and there was no significant difference between NT, TS, and CT at the filling stage.

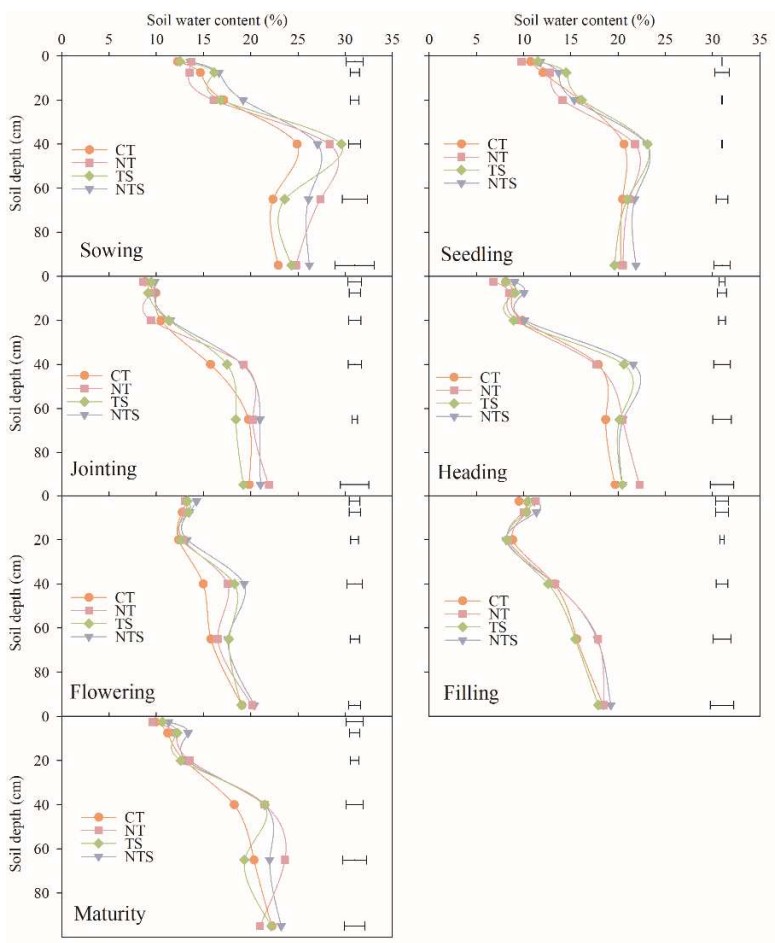

**Figure 2.** Soil water content at sowing, seedling, jointing, heading, flowering, filling, and maturity stages with the different tillage and stubble management practices (CT; NT; TS; NTS) in 2020–2022. CT, conventional tillage; NT, no tillage with no straw mulching; TS, conventional tillage with straw incorporation; NTS, no tillage with straw mulching; bars are LSD at $p < 0.05$. Same as below.

### 3.2. Stress-Related Substances

During 2020–2022, substances related to stress were significantly affected by tillage practices (Figure 3). Averaged across the three years, NTS and TS had significantly increased proline by 25.52% and 24.70% at the flowering stage, respectively. NTS, NT, and TS had significantly increased soluble protein by 2.69%, 2.08%, and 1.26%, respectively. The greatest differences in soluble protein content between treatments were found at the flowering stage, with NTS and NT significantly higher than CT by 4.70% and 5.76%, respectively. Similar responses of POD to tillage methods were observed at the seedling and jointing stages; the NTS, NT, and TS were significantly lower than CT by 13.72–30.56% compared to CT during 2020–2022. There was a significant increase in CAT content in NTS, NT, and TS compared to CT; NTS significantly increased CAT by 24.02% and 15.72% at the heading and flowering stages, respectively.

### 3.3. Biomass Accumulation, Grain Yield, and WUE

Tillage methods significantly affected wheat dry matter growth (Figure 4). In 2020–2022, the NTS treatment always maintained the highest biomass compared to the other treatments. The NTS treatment significantly increased spring wheat dry matter compared to CT at all growth stages. NTS increased spring wheat dry matter by 12.43–50.99% at the whole growth stage compared with CT. The grain yield was NTS > TS > NT > CT (Figure 5). NTS and TS significantly increased yield by 13.20–24.28% compared with CT in 2020–2022. The

NTS treatment had the highest WUE during 2020–2022, which was significantly higher than the other treatments.

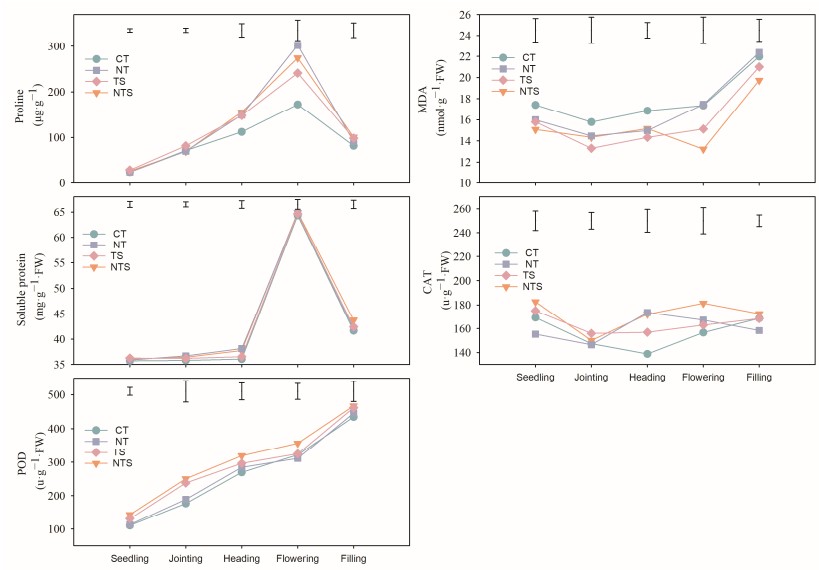

**Figure 3.** Stress-related substances of spring wheat at seedling, jointing, heading, flowering, and filling stages with the different tillage and stubble management practices (CT; NT; TS; NTS) in 2020–2022. CT, conventional tillage; NT, no tillage with no straw mulching; TS, conventional tillage with straw incorporation; NTS, no tillage with straw mulching; bars are LSD at *p* < 0.05. Same as below.

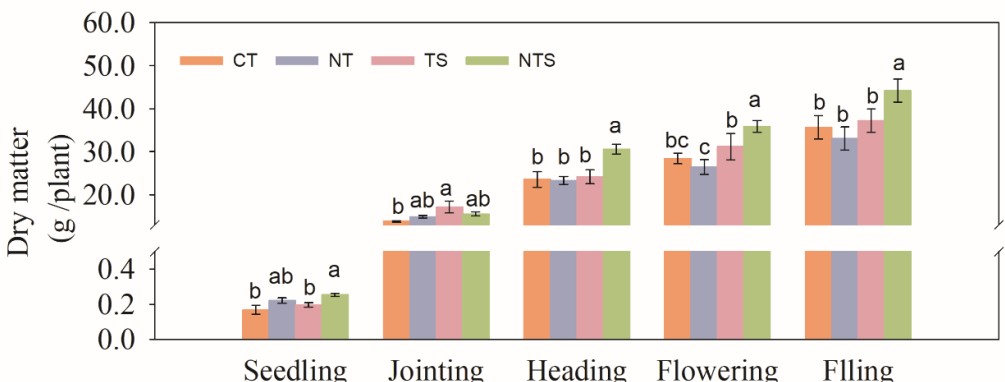

**Figure 4.** Effect of different stubble and management practices on dry matter accumulation in spring wheat at seedling, jointing, heading, flowering, and filling stages in 2020–2022. CT, conventional tillage; NT, no tillage with no straw mulching; TS, conventional tillage with straw incorporation; NTS, no tillage with straw mulching; means followed by different letters in a column are significantly different (*p* ≤ 0.05). Same as below.

*3.4. Correlations*

Significant correlations among the dry matter during growing season, grain yield, WUE and CAT, POD, MDA, soluble protein, and proline of spring wheat were observed (Table 2). The CAT, POD, proline, and soluble protein of spring wheat at the seedling stage were highly significant and positively associated with the grain yield. The dry matter at the jointing stage, grain, and WUE was positively associated with CAT; additionally, grain yield was significantly positively associated with POD and soluble protein. At the heading stage, dry matter and grain yield was positively associated with CAT, POD, and soluble protein. However, dry matter at the heading stage and grain yield had a negative correlation with MDA. The dry matter at the flowering stage and grain yield had a significantly positive

correlation with POD. The grain yield had a negative correlation with MDA. The WUE of spring wheat had a significantly positive correlation with CAT; in addition, the dry matter, grain yield had a significantly positive correlation with POD and soluble protein at the filling stage. The dry matter at the filling stage and grain yield was highly significant and negatively associated with MDA.

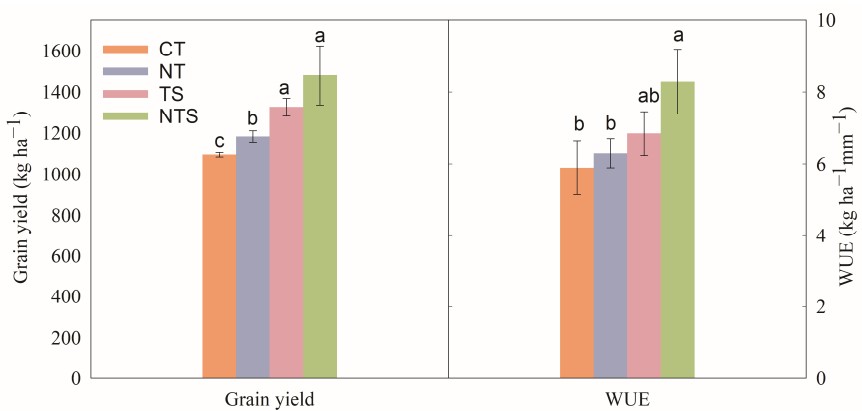

**Figure 5.** The grain yield and WUE of spring wheat under different methods in 2020–2022. CT, conventional tillage; NT, no tillage with no straw mulching; TS, conventional tillage with straw incorporation; NTS, no tillage with straw mulching; means followed by different letters in a column are significantly different ($p \leq 0.05$). Same as below.

**Table 2.** Correlation coefficient for correlations of dry matter at growing season, grain yield, WUE, and CAT, POD, MDA, soluble protein, proline across years for different growth stages of spring wheat.

| Growth stage | Index | CAT | POD | MDA | Soluble Protein | Proline |
|---|---|---|---|---|---|---|
| Seedling stage | Dry matter | −0.004 | −0.191 | 0.032 | −0.129 | 0.298 |
| | Grain yield | 0.917 ** | 0.904 ** | 0.900 ** | 0.928 ** | 0.919 ** |
| | WUE | 0.355 | 0.263 | 0.351 | −0.429 | 0.562 |
| Jointing stage | Dry matter | 0.783 ** | 0.121 | 0.500 | −0.523 | 0.460 |
| | Grain yield | 0.763 ** | 0.685 * | 0.402 | 0.924 ** | −0.424 |
| | WUE | 0.661 * | 0.158 | 0.421 | −0.483 | 0.381 |
| Heading stage | Dry matter | 0.707 * | 0.735 ** | −0.780 ** | 0.733 ** | 0.685 * |
| | Grain yield | 0.758 ** | 0.894 ** | −0.780 ** | 0.924 ** | −0.482 |
| | WUE | −0.041 | −0.268 | −0.170 | −0.391 | 0.376 |
| Flowering stage | Dry matter | 0.413 | 0.797 ** | −0.560 | −0.508 | 0.777 ** |
| | Grain yield | 0.730 ** | 0.898 ** | −0.755 ** | −0.271 | 0.767 ** |
| | WUE | 0.003 | 0.496 | 0.000 | 0.130 | 0.986 |
| Filling stage | Dry matter | 0.143 | 0.729 ** | −0.712 ** | 0.658 * | 0.647 * |
| | Grain yield | 0.249 | 0.909 ** | −0.926 ** | 0.880 ** | 0.775 ** |
| | WUE | 0.687 * | 0.267 | −0.608 * | −0.353 | 0.047 |

Note: *, significant at $p < 0.05$; **, significant at $p < 0.01$.

## 4. Discussion

### 4.1. Long-Term Conservation Tillage Optimizing Spring Wheat Yields by Improving Soil Physical and Chemistry

Many studies in the earlier part of this long-term locational experiment has found that the NTS treatment significantly reduced soil bulk density and increased soil nutrient content [31,38]. After more than 20 years of long-term trials later in this research, the NTS treatment caused soil bulk density to remain at the lowest level. Moreover, the NTS treatment had the highest content of total N, $NO_3^-$-N, and available P in the 0–30 cm soil layer; the pH was lower and more suitable for crop growth and development [10]. This

may be due to the increase in soil organic matter content by microbial decomposition as a result of returning straw to the field, which increases the soil nutrient content [39].

The application of no-till measures, in turn, maximizes the protection of soil voids for root growth [40], reduces the direct impact of rainfall directly on the soil, and retains water [38,41]. Soil water content is of great importance for crop growth and yield. This research found that the NTS and TS treatments significantly improved soil moisture content compared to that of CT, especially in the early crop growth period, and is in agreement with previous research [42,43]. This may be due to the improved physical properties of the soil and increased infiltration of precipitation by NTS compared to CT. In addition, straw mulching reduced soil evaporation, which is also consistent with previous studies [44,45]. In this study, the NT treatment soil moisture content was not significantly different from CT; however, no-till increased the dry matter accumulated and helped to use water more efficiently for spring wheat. This may be due to the fact that, in the no-till systems, plants accumulated more biomass, and the plants also extracted more water from the soil. Some studies have shown that NT can significantly improve soil moisture content [46]. This may be related to the local rainfall and other climatic conditions. One study found that, in the semi-arid zone of northern China, the TS treatment had no significant effect on soil water storage compared to that of CT [47]. On the one hand, it could be that the mulching prevents losing water through evaporation, but the plants used more water available in the soil, making it equal to the control treatment. On the other hand, it may also vary with the length of time when conservation tillage is implemented [48]. The availability of water and nutrients provides the basis for crop dry matter growth and yield improvement, further increasing the amount of return to the field, which creates a good cycle.

Over many years of research, soil's physicochemical properties have maintained the same pattern of response to different tillage treatments. In conclusion, long-term conservation tillage can improve the physical and chemical properties of the soil and store more water during the whole growth stage of the crop, thus providing a better soil environment for germination and growth of crops.

*4.2. No Tillage with Straw Mulching Improves Yield in Spring Wheat by Regulating Stress-Related Substances*

In our study, substances related to stress in spring wheat were significantly correlated with tillage practices (Figure 3). Changes in substances related to stress are a major component of the physiological mechanisms by which plants respond to external environmental stresses such as soil water deficiency [49,50]. Substances related to stress in the plant are important substances that affect the growth and yield of the plant [51]. External environmental stresses can inhibit crop growth and yield formation [52]. When plants are subjected to environmental stress, the balance of reactive oxygen metabolism is disrupted [53]. CAT regulates the level of hydrogen peroxide in plant cells, and different levels of drought stress can affect CAT content [54]. Under stress conditions, drought-tolerant plants tend to have higher POD activity than sensitive plants [55]. We found that CAT and POD contents in the NTS and TS treatments had stable and high levels throughout the whole growth stage. Moreover, POD and CAT contents were significantly and positively correlated with spring wheat yield and dry matter formation throughout the reproductive period of spring wheat (Table 2). This may be due to the synthesis of antioxidant enzymes that can eliminate or reduce the toxic effects of reactive oxygen species, thus reducing the damage caused by reactive oxygen species [56]. Similar findings were also seen in studies on rapeseed oil [57]. To prevent excessive water loss under external environmental stress, plants usually maintain cell proliferation and, thus, healthy vegetative development by reducing cytoplasmic osmosis or increasing the content of osmoregulatory substances [58]. Proline is a multifunctional molecule capable of preventing cellular damage by acting as an osmolyte and free radical (i.e., ROS) scavenger [59], and soluble proteins are among the molecular substances that make up cells [60]. In our study, we found that NTS-treated spring wheat had significantly lower MDA content and significantly higher proline and soluble protein

content than the other treatments. At the same time, dry matter accumulation and spring wheat yield were significantly negatively correlated with MDA and significantly positively correlated with proline and soluble protein. This is similar to previous studies, where MDA content was negatively correlated with crop growth indicators [61] and soluble protein was positively correlated with crop growth indicators [60].

In summary, in successive years of this study, spring wheat substances related to stress maintained the same response pattern to different tillage methods. Compared to the other treatments, the NTS treatment maintained higher levels of CAT, POD, proline, and soluble protein and the lowest MDA content though the whole growth stage of spring wheat. This may be an important way in which conservation tillage practices affect crop growth and yield.

## 5. Conclusions

No tillage with straw mulching improved soil's physicochemical properties and optimized dry matter accumulation, thus promoting yield. In addition, no tillage with straw mulching regulated substances related to stress in spring wheat, influenced water use-efficiency, and finally promoted grain yield. In summary, no tillage with straw mulching influences dry matter accumulation and water-use efficiency during the spring wheat whole growth stage by improving soil physicochemical properties and modulating spring wheat stress-tolerance substances, which in turn promoted yield formation.

**Author Contributions:** Conceptualization, L.L.; software, validation, L.L., C.D. and Z.E.; investigation, L.L.; resources, L.L.; data curation, C.D.; writing—original draft preparation, C.D.; writing—review and editing, J.X., L.L., Z.L. and L.W.; supervision, L.L. All authors have read and agreed to the published version of the manuscript.

**Funding:** This research was supported by the National Natural Science Foundation of China (32260549), the Fuxi Young Funds of Gansu Agricultural University (GAUfx-04Y09), and the Natural Science Foundation of Gansu Province (21JR7RA813).

**Data Availability Statement:** Not applicable.

**Conflicts of Interest:** The authors declare no conflict of interest.

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
