# Peer review of "Long-Term Conservation Tillage Increases Yield and Water Use Efficiency of Spring Wheat (Triticum aestivum L.) by Regulating Substances Related to Stress on the Semi-Arid Loess Plateau of China"

_agronomy, doi:10.3390/agronomy13051301_

Round 1

Reviewer 1 Report

Title suggestion: Implications for substitution straw mulching with tillage on yield and water use efficiency in spring wheat (Triticum aestivum L.) grown on the semi-arid Loess Plateau of China

Error in line 2   the sp. Name must be in small    aestivum

Line 54 your reference (1) It does not display the impact of erosion or plateau problems please replace with Specialist reference       look at line 525

حفظ الترجمة

Reference (2) in line 57 Important notice of the references used do not complete the sentences quoted from previous references Researchers should be careful in selecting specialized references   look at line 528

Line 150 حفظ الترجمة

The cultivar must be identified and its resistance to moisture stress must be determined

Line 151 What type of experimental design is used?

Line 218 replace spikes instead of ears

Line 220 replace spike instead of ear

The statistical analysis line 222 must use constant references, and even the program used must be proven by references

Table 1 The table must be self-identifying, even if abbreviations are mentioned in the research materials and methods

Clarify the time during 2020-2022 No need to add the standard error There is a number that is not followed by a letter

Table 2 write the unit of grain yield

Author Response

We thank the reviewers and editor for their valuable comments and suggestions to improve our manuscript. Our responses to each of the reviewers’ and editor’s comments are provided below:

Reviewer 1 comments:
Reviewer #1:

  1. Title suggestion: Implications for substitution straw mulching with tillage on yield and water use efficiency in spring wheat (Triticum aestivumL.) grown on the semi-arid Loess Plateau of China
  2. Error in line 2   the sp. Name must be in small aestivum
  3. Line 54 your reference (1) It does not display the impact of erosion or plateau problems please replace with Specialist reference look at line 525
  4. Reference (2) in line 57 Important notice of the references used do not complete the sentences quoted from previous references Researchers should be careful in selecting specialized referenceslook at line 528
  5. Line 150 The cultivar must be identified and its resistance to moisture stress must be determined
  6. Line 151 What type of experimental design is used?
  7. Line 218 replace spikes instead of ears
  8. Line 220 replace spike instead of ear
  9. The statistical analysis line 222 must use constant references, and even the program used must be proven by references
  10. Table 1 The table must be self-identifying, even if abbreviations are mentioned in the research materials and methods

Clarify the time during 2020-2022 No need to add the standard error There is a number that is not followed by a letter

  1. Table 2write the unit of grain yield
  2. General comments: Conservationist agriculture is well established, and there is limited novelty in the observation that no tillage protects the soil, improves the use of water, and influences the substances related to stress. Assuming that previous knowledge, the authors should propose a clear scientific hypothesis to justify what is the novelty of this present study. The authors should express clearly that this manuscript is part of a long-term study. This detail changes the understanding of the study. It is just mentioned very quickly, but it deserves more attention in the Introduction, with a few lines about the previous findings of the experiment. Highlighting the novelty is even more important because it is part of a long-term study, and it seems that all the results are similar to previous reports.

-1. Error in line 2   the sp. Name must be in small aestivum

Answer: In the latest version of the revision we have rewritten the abstract section, "name has be in small aestivum".

-2. Line 54 your reference (1) It does not display the impact of erosion or plateau problems please replace with Specialist reference look at line 525

Answer: We have made changes to the relevant literature and checked other literature.

-3. Reference (2) in line 57 Important notice of the references used do not complete the sentences quoted from previous references Researchers should be careful in selecting specialized references look at line 528

Answer: We have made changes to the relevant literature and checked other literature.

-4. Line 150 المThe cultivar must be identified and its resistance to moisture stress must be determined

Answer: We have revised it. Specifically: "Dingxi 40" is a national appraisal variety (National Appraisal Wheat 2009032), with medium drought resistance.

-5. Line 151 What type of experimental design is used?

Answer: The experiment was a randomised block design with four treatments and three replications of each treatment.

-6. Line 218 replace spikes instead of ears Line 220 replace spike instead of ear

Answer: We have amended it.

-7. The statistical analysis line 222 must use constant references, and even the program used must be proven by references

Answer: In the latest revision, we have rewritten the statistical analysis section.

-8. Table 1 The table must be self-identifying, even if abbreviations are mentioned in the research materials and methods

Answer: In the latest revision, we have redrawn Table 1. And the contents of the table are marked and explained.

-9. Clarify the time during 2020-2022 No need to add the standard error There is a number that is not followed by a letter

Answer: We have amended it.

-10. Table 2 write the unit of grain yield

Answer: We have amended it.

-11. General comments: Conservationist agriculture is well established, and there is limited novelty in the observation that no tillage protects the soil,  improves the use of water, and influences the substances related to stress. Assuming that previous knowledge, the authors should propose a clear scientific hypothesis to justify what is the novelty of this present study. The authors should express clearly that this manuscript is part of a long-term study. This detail changes the understanding of the study. It is just mentioned very quickly, but it deserves more attention in the Introduction, with a few lines about the previous findings of the experiment. Highlighting the novelty is even more important because it is part of a long-term study, and it seems that all the results are similar to previous reports.

Answer: In the latest revised edition, we have reorganised and rewritten the Preface and the Discussion and Conclusion sections to ensure the novelty of the article's content.

Reviewer 2 Report

General: The content and overall structure of the manuscript are good. The manuscript possesses promising findings, primarily to address the use of alternative soil management practices on spring wheat on the semi-arid Loess Plateau of China. However, there are comments and observations indicated in the pdf manuscript. The author can consider addressing them in the manuscript. How and why is this work novel and different from others? In the introduction, articulate how this research contributes to new knowledge in the discipline. The specific comments are as follows:

Specific comments:

The introduction section lacks coherence. The author can describe some previous findings of soil management on crop/wheat’s antioxidant enzymes and cross-regulation substances. The experiment design was RCB or CRD? Need to be clearly mentioned. The authors indicated “….and a double sequence rotation cropping pattern of spring wheat and peas with interannual rotation of the two crops.” Is this also considered a factor/treatment? If not need proper clarification on it. Whether the trial was irrigated or rainfed needs to define. How many soil samples for soil moisture contents study were taken per 80 m2 unit plot, and the two methods were indicated in two places. If these are steps, explain them accordingly. 

Experimental title, and in Table 6, water use efficiency (WUE) was reported. However, there was data on WUE neither recorded nor shown other than soil moisture monitoring data at different growth stages of wheat. The authors, in some places, used drought indices data. Not sure what data they were. These need to be clearly explained.                                                                                      

The subheadings of the discussion section themselves are findings. Not sure if this is the proper way to present here. However, the whole discussion section looks like a repetition of results. This section should be appropriately interpreted. The results should be discussed in relation to the previous findings. The manicurist has a lot of promising conclusions. The authors can consider rewording to avoid repetition.

Author Response

We thank the reviewers and editor for their valuable comments and suggestions to improve our manuscript. Our responses to each of the reviewers’ and editor’s comments are provided below:

Reviewer 2 comments:

General: The content and overall structure of the manuscript are good. The manuscript possesses promising findings, primarily to address the use of alternative soil management practices on spring wheat on the semi-arid Loess Plateau of China. However, there are comments and observations indicated in the pdf manuscript. The author can consider addressing them in the manuscript. How and why is this work novel and different from others? In the introduction, articulate how this research contributes to new knowledge in the discipline. The specific comments are as follows:

Specific comments:

The introduction section lacks coherence. The author can describe some previous findings of soil management on crop/wheat’s antioxidant enzymes and cross-regulation substances. The experiment design was RCB or CRD? Need to be clearly mentioned. The authors indicated “….and a double sequence rotation cropping pattern of spring wheat and peas with interannual rotation of the two crops.” Is this also considered a factor/treatment? If not need proper clarification on it. Whether the trial was irrigated or rainfed needs to define. How many soil samples for soil moisture contents study were taken per 80 m2 unit plot, and the two methods were indicated in two places. If these are steps, explain them accordingly.

Experimental title, and in Table 6, water use efficiency (WUE) was reported. However, there was data on WUE neither recorded nor shown other than soil moisture monitoring data at different growth stages of wheat. The authors, in some places, used drought indices data. Not sure what data they were. These need to be clearly explained.

 The subheadings of the discussion section themselves are findings. Not sure if this is the proper way to present here. However, the whole discussion section looks like a repetition of results. This section should be appropriately interpreted. The results should be discussed in relation to the previous findings. The manicurist has a lot of promising conclusions. The authors can consider rewording to avoid repetition.

  1. The introduction section lacks coherence. The author can describe some previous findings of soil management on crop/wheat’s antioxidant enzymes and cross-regulation substances.

Answer: Thank you very much for your valuable suggestions. In the latest revised version of the manuscript, we have rewritten the introduction section in line with your comments.

  1. The experiment design was RCB or CRD? Need to be clearly mentioned.

Answer: In the latest revised version of the manuscript, we have rewritten the experimental design section in response to your comments.

  1. The authors indicated “….and a double sequence rotation cropping pattern of spring wheat and peas with interannual rotation of the two crops.” Is this also considered a factor/treatment?

Answer: Spring wheat and pea inter-annual rotations are an important component of long-term conservation tillage trials, but are not treated as a factor in our manuscript.

  1. Whether the trial was irrigated or rainfed needs to define. How many soil samples for soil moisture contents study were taken per 80 m2 unit plot, and the two methods were indicated in two places.

Answer: The test area is a rain-fed agricultural area and no irrigation treatment was required for this trial. Soil samples for soil moisture and physical and chemical properties were taken by soil auger.

  1. Experimental title, and in Table 6, water use efficiency (WUE) was reported. However, there was data on WUE neither recorded nor shown other than soil moisture monitoring data at different growth stages of wheat. The authors, in some places, used drought indices data. Not sure what data they were. These need to be clearly explained.

Answer: We have recompiled the article data in the latest revision and have rewritten the analysis of the results accordingly.

  1. The subheadings of the discussion section themselves are findings. Not sure if this is the proper way to present here. However, the whole discussion section looks like a repetition of results. This section should be appropriately interpreted. The results should be discussed in relation to the previous findings. The manicurist has a lot of promising conclusions. The authors can consider rewording to avoid repetition.

Answer: We have reorganised the article and rewritten the discussion and conclusion sections. Thank you for your suggestions.

Reviewer 3 Report

General comments: Conservationist agriculture is well established, and there is limited novelty in the observation that no tillage protects the soil,  improves the use of water, and influences the substances related to stress. Assuming that previous knowledge, the authors should propose a clear scientific hypothesis to justify what is the novelty of this present study.

The authors should express clearly that this manuscript is part of a long-term study. This detail changes the understanding of the study. It is just mentioned very quickly, but it deserves more attention in the Introduction, with a few lines about the previous findings of the experiment. Highlighting the novelty is even more important because it is part of a long-term study, and it seems that all the results are similar to previous reports.

The title is not expressing adequately the content of the study. While the title highlights yield and water use efficiency, most of the results presented in the manuscript are about the substances related to stress.

Abstract:

In the first sentence of the Abstract, I suggest that the authors widen the reach of this study. Water scarcity is a detrimental factor not only for wheat in the Loess Plateau, but for most of the agriculture in the world. Then the suggestion is to state something like “Water scarcity is the most important constraint for agriculture in the world”.

The abstract is too short on the description of the methods. It is necessary at least a brief description of some methods to make possible to understand the results.

Line 20: this sentence can be read as “soil nutrition”; maybe you could change for “soil moisture and plant nutritrion”.

Lines 23-24: The year 2020 was wet (annual… 292 mm on the… (please correct the same thing in the following sentence as well).

Line 27: This sentence needs to be reformulated to be clear (NTS significantly improved soil moisture and nutrient.) The sentence is vague and can be interpreted like if the treatment increased water and nutrients. Maybe the treatment conserved the water and nutrients. A suggestion to improve is: “NTS improved the efficiency of water and nutrients”.

Lines 28-29: It is not informed what increased in the comparison between NTS and CT.

Line 30: “cross-regulation substance” is not a clear concept. Please rephrase it with clear words and revise the whole manuscript to improve the term. I suppose you mean “stress-related substances”.

Please reconsider if you will compare the “years”; maybe the study is only on the soil treatments, and the years are random variation. If you decide to consider the years as a treatment, the manuscript needs to be fully restructured. In that case, the treatments should be assumed as a factorial design, statistical analysis and discussion should be changed accordingly. My opinion is that the study should consider only the soil treatments, which are the really meaningful objective. Keep the years just as replications on time.

Between lines 31 and 38, the presentation of results is very complicated (comparing each substance that increased or decreased in each year). Because of that, the discussion is too long on the substances related to stress varying each year. At the end the reader cannot grasp any relevant information because it is too complicated. The occasional up and downs among years are just noisance and should not be detailed. Only the average across years should be considered.

If the comparison were always made with the CT treatment, it would be easier to describe that treatment as the control, and just make the comparison without the need to explain repeatedly.

Key words need to be improved: the scientific name of wheat is repeated from the title. Please replace “tillage measures” (the meaning of these terms is not clear). “Cross-regulation substances” is not an established concept – as they are not in the title, you can mention them as key words. WUE is repeated from the title.

Line 63: Maybe instead of “tillage measures” you mean “tillage methods” (revise that in the whole manuscript). Replace “reduce” with “reduced”.

Line 80: …reduces soil bulk density.

Methods:

More details are needed on how the experiment was managed. Although the description is cited from another study, it needs to be at least briefly discussed.

- What is the planting time for wheat and pea?

- How was the straw provided in the NTS treatment and how was it removed in CT and NT treatments?

- Lines 167-169: It seems that this phrase lost a half; it lacks a verb and something else.

- Line 211: please rephrase this sentence. I could not understand what means repeating a plant three times.

- Line 216: instead of yielded do you mean “weighed”?

- Line 236: replace “showed” with “they were” (you need to correct all the uses of the verb show in the following sentences. They are all inadequate).

- Line 242: correct the typos “availabl”

Statistical analysis is not detailed enough. Were the years considered a treatment? As the results of each year were presented separately in table 2, it seems that the year was considered a treatment (instead of a random factor). As previously discussed, this problem needs to be solved. In the current version, I assume that the statistical analysis is not right.

Results

Again, I suggest that the data should be presented as an average across years. The only relevant meaning of the treatments is its effect across years at varying environmental conditions. For instance, the soil moisture content should be influenced by treatments in any condition (i.e., rainy or dry year); if the result is not repeated in contrasting conditions, then the treatment is not consistently influencing that characteristic and that should be the conclusion of the study. In the way it is presented, the results, figures, and tables are very complicated and difficult to understand. If the results were pooled, the reader could understand the relevant information. This same reasoning should be considered for all the measurements in this study.

Line 333: correct the typo “accsumption”

Figure 4: The type of figure is not adequate. You cannot stack the values of biomass at different developmental phases. The horizontal axis should be the time (developmental phases), and the values of biomass should be pooled (average of 3 years) and one line for each treatment.

The details of yield components are very difficult to show any relevant information to the reader. This is another case in which showing the yield components does not add any useful information. You could just present the final grain yield, and the reader would not lose any relevant information. Unless the authors offer a reason (i.e., discuss the result extracting some useful information), you should present only the data on grain yield and WUE.

Please double check if the unit for WUE has should include the area. I think it should be only kg/mm.

Line 360: the subsection is “Correlations”, but the content is a Principal Components Analysis (PCA). Please correct the title of the subsection. This analysis was not explained in the section for Statistical Analysis. This PCA analysis does not add any useful information to this study. The paragraph 367-372 is completely wrong. The scores cannot be combined, and they do not express anything to support that some treatments were better than other. How could you pool the coefficients and apply a mean comparison test among them? This is completely wrong! This is just a clustering analysis, and the authors did not use it properly.

The discussion of results is not discussing the current study, but it is a general discussion on the benefits of no-tillage system.

Author Response

We thank the reviewers and editor for their valuable comments and suggestions to improve our manuscript. Our responses to each of the reviewers’ and editor’s comments are provided below:

Reviewer 3 comments:

  1. The title is not expressing adequately the content of the study. While the title highlights yield and water use efficiency, most of the results presented in the manuscript are about the substances related to stress.
  2. Abstract: In the first sentence of the Abstract, I suggest that the authors widen the reach of this study. Water scarcity is a detrimental factor not only for wheat in the Loess Plateau, but for most of the agriculture in the world. Then the suggestion is to state something like “Water scarcity is the most important constraint for agriculture in the world”.
  3. The abstract is too short on the description of the methods. It is necessary at least a brief description of some methods to make possible to understand the results.
  4. Line 20: this sentence can be read as “soil nutrition”; maybe you could change for “soil moisture and plant nutritrion”.
  5. Lines 23-24: The year 2020 was wet (annual… 292 mm on the… (please correct the same thing in the following sentence as well).
  6. Line 27: This sentence needs to be reformulated to be clear (NTS significantly improved soil moisture and nutrient.) The sentence is vague and can be interpreted like if the treatment increased water and nutrients. Maybe the treatment conserved the water and nutrients. A suggestion to improve is: “NTS improved the efficiency of water and nutrients”.
  7. Lines 28-29: It is not informed what increased in the comparison between NTS and CT.
  8. Line 30: “cross-regulation substance” is not a clear concept. Please rephrase it with clear words and revise the whole manuscript to improve the term. I suppose you mean “stress-related substances”.
  9. Please reconsider if you will compare the “years”; maybe the study is only on the soil treatments, and the years are random variation. If you decide to consider the years as a treatment, the manuscript needs to be fully restructured. In that case, the treatments should be assumed as a factorial design, statistical analysis and discussion should be changed accordingly. My opinion is that the study should consider only the soil treatments, which are the really meaningful objective. Keep the years just as replications on time.
  10. Between lines 31 and 38, the presentation of results is very complicated (comparing each substance that increased or decreased in each year). Because of that, the discussion is too long on the substances related to stress varying each year. At the end the reader cannot grasp any relevant information because it is too complicated. The occasional up and downs among years are just noisance and should not be detailed. Only the average across years should be considered.
  11. If the comparison were always made with the CT treatment, it would be easier to describe that treatment as the control, and just make the comparison without the need to explain repeatedly.
  12. Key words need to be improved: the scientific name of wheat is repeated from the title. Please replace “tillage measures” (the meaning of these terms is not clear). “Cross-regulation substances” is not an established concept – as they are not in the title, you can mention them as key words. WUE is repeated from the title.
  13. Line 63: Maybe instead of “tillage measures” you mean “tillage methods” (revise that in the whole manuscript). Replace “reduce” with “reduced”.
  14. Line 80: …reduces soil bulk density.
  15. Methods:

More details are needed on how the experiment was managed. Although the description is cited from another study, it needs to be at least briefly discussed.

  1. - What is the planting time for wheat and pea?
  2. - How was the straw provided in the NTS treatment and how was it removed in CT and NT treatments?
  3. - Lines 167-169: It seems that this phrase lost a half; it lacks a verb and something else.
  4. - Line 211: please rephrase this sentence. I could not understand what means repeating a plant three times.
  5. - Line 216: instead of yielded do you mean “weighed”?
  6. - Line 236: replace “showed” with “they were” (you need to correct all the uses of the verb show in the following sentences. They are all inadequate).
  7. - Line 242: correct the typos “availabl”
  8. Statistical analysis is not detailed enough. Were the years considered a treatment? As the results of each year were presented separately in table 2, it seems that the year was considered a treatment (instead of a random factor). As previously discussed, this problem needs to be solved. In the current version, I assume that the statistical analysis is not right.
  9. Results

Again, I suggest that the data should be presented as an average across years. The only relevant meaning of the treatments is its effect across years at varying environmental conditions. For instance, the soil moisture content should be influenced by treatments in any condition (i.e., rainy or dry year); if the result is not repeated in contrasting conditions, then the treatment is not consistently influencing that characteristic and that should be the conclusion of the study. In the way it is presented, the results, figures, and tables are very complicated and difficult to understand. If the results were pooled, the reader could understand the relevant information. This same reasoning should be considered for all the measurements in this study.

  1. Line 333: correct the typo “accsumption”
  2. Figure 4: The type of figure is not adequate. You cannot stack the values of biomass at different developmental phases. The horizontal axis should be the time (developmental phases), and the values of biomass should be pooled (average of 3 years) and one line for each treatment.
  3. The details of yield components are very difficult to show any relevant information to the reader. This is another case in which showing the yield components does not add any useful information. You could just present the final grain yield, and the reader would not lose any relevant information. Unless the authors offer a reason (i.e., discuss the result extracting some useful information), you should present only the data on grain yield and WUE.
  4. Please double check if the unit for WUE has should include the area. I think it should be only kg/mm.
  5. Line 360: the subsection is “Correlations”, but the content is a Principal Components Analysis (PCA). Please correct the title of the subsection. This analysis was not explained in the section for Statistical Analysis. This PCA analysis does not add any useful information to this study. The paragraph 367-372 is completely wrong. The scores cannot be combined, and they do not express anything to support that some treatments were better than other. How could you pool the coefficients and apply a mean comparison test among them? This is completely wrong! This is just a clustering analysis, and the authors did not use it properly.一个聚类分析,作者没有正确30. The discussion of results is not discussing the current study, but it is a general discussion on the benefits of no-tillage system.
  6. The title is not expressing adequately the content of the study. While the title highlights yield and water use efficiency, most of the results presented in the manuscript are about the substances related to stress.

  1. The title is not expressing adequately the content of the study. While the title highlights yield and water use efficiency, most of the results presented in the manuscript are about the substances related to stress.

Answer: The title of the article was amended to read" Long term conservation tillage increases yield and water use efficiency of spring wheat (Triticum aestivum L.) by regulating stress-related substances on the semi-arid Loess Plateau of China"

  1. Abstract: In the first sentence of the Abstract, I suggest that the authors widen the reach of this study. Water scarcity is a detrimental factor not only for wheat in the Loess Plateau, but for most of the agriculture in the world. Then the suggestion is to state something like “Water scarcity is the most important constraint for agriculture in the world”.

Answer: In the latest revision, we have modified it.

  1. The abstract is too short on the description of the methods. It is necessary at least a brief description of some methods to make possible to understand the results.

Answer: In the latest revision, we have rewritten the abstract section.

  1. Line 20: this sentence can be read as “soil nutrition”; maybe you could change for “soil moisture and plant nutritrion”.

Answer: In the latest revision, we have modified it.

  1. Lines 23-24: The year 2020 was wet (annual… 292 mm on the… (please correct the same thing in the following sentence as well).

Answer: In the latest revision, we have modified it.

  1. Line 27: This sentence needs to be reformulated to be clear (NTS significantly improved soil moisture and nutrient.) The sentence is vague and can be interpreted like if the treatment increased water and nutrients. Maybe the treatment conserved the water and nutrients. A suggestion to improve is: “NTS improved the efficiency of water and nutrients”.

Answer: In the latest revision, we have modified it.

  1. Lines 28-29: It is not informed what increased in the comparison between NTS and CT.

Answer: In the latest revision, we have rewritten the abstract section. In which the content has been changed. Specifically: The NTS treatment had the lowest soil bulk and pH and the highest levels of total N, NO3--N and available P.

  1. Line 30: “cross-regulation substance” is not a clear concept. Please rephrase it with clear words and revise the whole manuscript to improve the term. I suppose you mean “stress-related substances”.

Answer: In the latest revision, we have modified it.

  1. Please reconsider if you will compare the “years”; maybe the study is only on the soil treatments, and the years are random variation. If you decide to consider the years as a treatment, the manuscript needs to be fully restructured. In that case, the treatments should be assumed as a factorial design, statistical analysis and discussion should be changed accordingly. My opinion is that the study should consider only the soil treatments, which are the really meaningful objective. Keep the years just as replications on time.

Answer: In the latest revision, we have considered factors such as year and rainfall as random factors, and tillage practices as the only fixed factors, and have integrated and re-analysed the data for three years.

  1. Between lines 31 and 38, the presentation of results is very complicated (comparing each substance that increased or decreased in each year). Because of that, the discussion is too long on the substances related to stress varying each year. At the end the reader cannot grasp any relevant information because it is too complicated. The occasional up and downs among years are just noisance and should not be detailed. Only the average across years should be considered.

Answer: In the latest revision, we have rewritten the abstract section.

  1. If the comparison were always made with the CT treatment, it would be easier to describe that treatment as the control, and just make the comparison without the need to explain repeatedly.

Answer: In the latest revision, we have rewritten the abstract section.

  1. Key words need to be improved: the scientific name of wheat is repeated from the title. Please replace “tillage measures” (the meaning of these terms is not clear). “Cross-regulation substances” is not an established concept – as they are not in the title, you can mention them as key words. WUE is repeated from the title.

Answer: In the latest revision, we have rewritten the keywords. Specifically: spring wheat; tillage methods; stress-related substances; yield;

  1. Line 63: Maybe instead of “tillage measures” you mean “tillage methods” (revise that in the whole manuscript). Replace “reduce” with “reduced”.

Answer: In the latest revision, we have modified it.

  1. Line 80: …reduces soil bulk density.

Answer: In the latest revision, we have modified it.

  1. Methods:

More details are needed on how the experiment was managed. Although the description is cited from another study, it needs to be at least briefly discussed.

Answer: In the latest revision, we have rewritten the s-test design section to provide a full description of the experiment.

  1. - What is the planting time for wheat and pea?

Answer: In the latest revision, we have rewritten the design section to provide a full description of the experiment. Specifically: Spring wheat was planted in late March and harvested in late July each year; peas were planted in early April and harvested in early July each year.

  1. - How was the straw provided in the NTS treatment and how was it removed in CT and NT treatments?

Answer: In the latest revision, we have rewritten the design section to provide a full description of the experiment. Specifically: no tillage with no straw mulching (NT), the field was left un-tilled throughout the experiment. Sowing and fertilization were performed with seedling-machine at the same time; and conventional tillage with straw incorporation (TS), the field was ploughed and harrowed exactly as CT treatment, but with straw incorporated at the first plough.

  1. - Lines 167-169: It seems that this phrase lost a half; it lacks a verb and something else.

Answer: In the latest revision, we have rewritten it. Specifically: All samples for this study were collected in the spring wheat field. Weeds were removed by hand during the growing season and controlled with herbicides during the fallow period.

  1. - Line 211: please rephrase this sentence. I could not understand what means repeating a plant three times.

Answer: In the latest revision, we have rewritten it. Specifically: Fifteen spring wheat leaves were randomly taken in each plot, wrapped in aluminum foil and immediately placed in a bubble box containing liquid nitrogen, which was brought back to the laboratory and transferred to a -80 ℃ refrigerator for measurement.

  1. - Line 216: instead of yielded do you mean “weighed”?

Answer: In the latest revision, we have modified it.

  1. - Line 236: replace “showed” with “they were” (you need to correct all the uses of the verb show in the following sentences. They are all inadequate).

Answer: In the latest revision, we have modified it.

  1. - Line 242: correct the typos “availabl”

Answer: In the latest revision, we have modified it.

  1. Statistical analysis is not detailed enough. Were the years considered a treatment? As the results of each year were presented separately in table 2, it seems that the year was considered a treatment (instead of a random factor). As previously discussed, this problem needs to be solved. In the current version, I assume that the statistical analysis is not right.

Answer: In the latest revision, we have considered factors such as year and rainfall as random factors, and tillage practices as the only fixed factors, and have integrated and re-analysed the data for three years.

  1. Results

Again, I suggest that the data should be presented as an average across years. The only relevant meaning of the treatments is its effect across years at varying environmental conditions. For instance, the soil moisture content should be influenced by treatments in any condition (i.e., rainy or dry year); if the result is not repeated in contrasting conditions, then the treatment is not consistently influencing that characteristic and that should be the conclusion of the study. In the way it is presented, the results, figures, and tables are very complicated and difficult to understand. If the results were pooled, the reader could understand the relevant information. This same reasoning should be considered for all the measurements in this study.

Answer: In the latest revision, we integrate the data for analysis.

  1. Line 333: correct the typo “accsumption”

Answer: In the latest revision, we have modified it.

  1. Figure 4: The type of figure is not adequate. You cannot stack the values of biomass at different developmental phases. The horizontal axis should be the time (developmental phases), and the values of biomass should be pooled (average of 3 years) and one line for each treatment.

Answer: In the latest revision, we have redrawn the relevant fig.

  1. The details of yield components are very difficult to show any relevant information to the reader. This is another case in which showing the yield components does not add any useful information. You could just present the final grain yield, and the reader would not lose any relevant information. Unless the authors offer a reason (i.e., discuss the result extracting some useful information), you should present only the data on grain yield and WUE.

Answer: In the latest revised draft, we have revised the content in response to your comments.

  1. Please double check if the unit for WUE has should include the area. I think it should be only kg/mm.

Answer: In the latest revision, we have modified it.

  1. Line 360: the subsection is “Correlations”, but the content is a Principal Components Analysis (PCA). Please correct the title of the subsection. This analysis was not explained in the section for Statistical Analysis. This PCA analysis does not add any useful information to this study. The paragraph 367-372 is completely wrong. The scores cannot be combined, and they do not express anything to support that some treatments were better than other. How could you pool the coefficients and apply a mean comparison test among them? This is completely wrong! This is just a clustering analysis, and the authors did not use it properly.

Answer: In the latest revision, we have reorganised the results section and analysed it.

  1. The discussion of results is not discussing the current study, but it is a general discussion on the benefits of no-tillage system.

Answer: In the latest revision, we have rewritten the discussion section.

  1. The title is not expressing adequately the content of the study. While the title highlights yield and water use efficiency, most of the results presented in the manuscript are about the substances related to stress.

Answer: The title of the article was amended to read" Long term conservation tillage increases yield and water use efficiency of spring wheat (Triticum aestivum L.) by regulating stress-related substances on the semi-arid Loess Plateau of China"

Round 2

Reviewer 3 Report

This manuscript was significantly revised and improved compared to the previous version. I just suggest some correction to improve some details.

Abstract

Line 36: correct the typo “cross”

Introduction

Line 48: Write the scientific name in italics

Line 52: revise the whole manuscript for typos and corrections. For instance, in this line there are spaces missing between words.

Lines 65-66: Please rephrase this sentence. There is not “stress-resistant substances”. Maybe you mean “substances related to stress” (revise all the manuscript because this wrong expression is used multiple times). There is not “ecological balance” inside a plant. The term ecological implies the relation among many species and the environment. Please correct it.

Line 77: Use subscript letters for the number in the chemical formula (revise the same typo in the whole manuscript).

Methods

Lines 208-209: Just correct the words that are in bold font.

Results

Table 1: As there is space in the columns, please write the treatments in full instead of using the abbreviations (CT, TS, NTS etc). You can make the reading easier to the reader when you avoid abbreviations.

Discussion:

Line 322: It is not mandatory that the no-till increase water content in the soil. In fact, it helps to use the water more efficiently. See that the plants accumulated more biomass in the no-till system, and then the plants also extracted more water from the soil. The mulching prevents losing water through evaporation, but the plants take advantage and use more water available in the soil, making it equal to the control treatment. Maybe you could discuss a little more that issue in the manuscript.

Line 386: Please rewrite this sentence because I could understand what you mean: “…higher live maximum levels of…”

Conclusions

Line 401: correct the verb “…in turn promoted yield…”

Author Response

Dr. Editor

Response for manuscript agronomy-2302046 Long-term conservation tillage increases yield and water use efficiency of spring wheat (Triticum aestivum L.) by regulating substances related to stress on the semi-arid Loess Plateau of China

Thanks for providing us with this great opportunity to submit a revised version of our manuscript. We appreciate the detailed and constructive comments provided by the reviewers. We have carefully revised the manuscript by incorporating all the suggestions by the review panel.

We hope this revised manuscript has addressed your concerns, and look forward to hearing from you.

Sincerely,

The Authors

Encl. Responses to the comments from Reviewer 3.

We thank the reviewers and editor for their valuable comments and suggestions to improve our manuscript. Our responses to each of the reviewers’ and editor’s comments are provided below:

Reviewer 3 comments:
Reviewer #3:

This manuscript was significantly revised and improved compared to the previous version. I just suggest some correction to improve some details.

Abstract

  1. Line 36: correct the typo “cross”

Introduction

  1. Line 48: Write the scientific name in italics
  2. Line 52: revise the whole manuscript for typos and corrections. For instance, in this line there are spaces missing between words.
  3. Lines 65-66: Please rephrase this sentence. There is not “stress-resistant substances”. Maybe you mean “substances related to stress” (revise all the manuscript because this wrong expression is used multiple times). There is not “ecological balance” inside a plant. The term ecological implies the relation among many species and the environment. Please correct it.
  4. Line 77: Use subscript letters for the number in the chemical formula (revise the same typo in the whole manuscript).

Methods

  1. Lines 208-209: Just correct the words that are in bold font.

Results

  1. Table 1: As there is space in the columns, please write the treatments in full instead of using the abbreviations (CT, TS, NTS etc). You can make the reading easier to the reader when you avoid abbreviations.

Discussion:

  1. Line 322: It is not mandatory that the no-till increase water content in the soil. In fact, it helps to use the water more efficiently. See that the plants accumulated more biomass in the no-till system, and then the plants also extracted more water from the soil. The mulching prevents losing water through evaporation, but the plants take advantage and use more water available in the soil, making it equal to the control treatment. Maybe you could discuss a little more that issue in the manuscript.
  2. Line 386: Please rewrite this sentence because I couldunderstand what you mean: “…higher live maximum levels of…”

Conclusions

  1. Line 401: correct the verb “…in turn promoted yield…”

  1. Line 36: correct the typo “cross”

Answer: We have modified it in the latest revision.

  1. Line 48: Write the scientific name in italics

Answer: We have modified it in the latest revision.

  1. Line 52: revise the whole manuscript for typos and corrections. For instance, in this line there are spaces missing between words.

Answer: We have modified it in the latest revision. We also double-checked the grammar and vocabulary and other issues throughout the text to ensure that there are no l similar problems.

  1. Lines 65-66: Please rephrase this sentence. There is not “stress-resistant substances”. Maybe you mean “substances related to stress” (revise all the manuscript because this wrong expression is used multiple times). There is not “ecological balance” inside a plant. The term ecological implies the relation among many species and the environment. Please correct it.

Answer: We have modified it in the latest revision. We have corrected the relevant expressions in the whole article.

  1. Line 77: Use subscript letters for the number in the chemical formula (revise the same typo in the whole manuscript).

Answer: We have modified it in the latest revision. We also double-checked the grammar and vocabulary and other issues throughout the text to ensure that there are no l similar problems.

  1. Lines 208-209: Just correct the words that are in bold font.

Answer: We have modified it in the latest revision. We also double-checked the grammar and vocabulary and other issues throughout the text to ensure that there are no l similar problems.

  1. Table 1: As there is space in the columns, please write the treatments in full instead of using the abbreviations (CT, TS, NTS etc). You can make the reading easier to the reader when you avoid abbreviations.

Answer: Thanks for your suggestion, we have changed the form.

  1. Line 322: It is not mandatory that the no-till increase water content in the soil. In fact, it helps to use the water more efficiently. See that the plants accumulated more biomass in the no-till system, and then the plants also extracted more water from the soil. The mulching prevents losing water through evaporation, but the plants take advantage and use more water available in the soil, making it equal to the control treatment. Maybe you could discuss a little more that issue in the manuscript.

Answer: Thank you very much for helping us to open our minds. We have added the relevant content to the discussion section in the latest revision, but we are sorry and regret that we could not find the relevant article you published and cited it.

  1. Line 386: Please rewrite this sentence because I couldunderstand what you mean: “…higher live maximum levels of…”

Answer: Thanks for your suggestion, we have changed the sentence.

  1. Line 401: correct the verb “…in turn promoted yield…”

Answer: Thanks for your suggestion, we have changed the sentence.
